# Linking Entrepreneurship to Productivity: Using a Composite Indicator for Farm-Level Innovation in UK Agriculture with Secondary Data

Yiorgos Gadanakis *, Jorge Campos-González and Philip Jones

School of Agriculture, Policy and Development, University of Reading, Reading RG6 6AR, UK;
jorge.camposgonzalez@reading.ac.uk (J.C.-G.); p.jones2@reading.ac.uk (P.J.)
* Correspondence: g.gadanakis@reading.ac.uk

**Abstract:** In agriculture, the intricate relationship between innovation, productivity, and entrepreneurship is underexplored. Despite the widely recognized role of innovation in driving productivity, concrete indicators and comprehensive farm-level studies are lacking. This research aims to unravel this complexity by exploring the impact of innovation, specifically in agricultural entrepreneurship, on transformative changes in farm productivity. The work presented in this manuscript explores how farm-level data derived from the Farm Business Survey (FBS) for the period between 2003 and 2014 is used to identify innovators and to assesses changes in productivity, technical efficiency, and economic efficiency. Therefore, it aims to contribute to comprehensively exploring the role of innovation, particularly within the context of entrepreneurship in agriculture, and its influence on driving transformative changes in farm productivity. Results reveal significant productivity variation and a moderate overall improvement. Furthermore, investment in human resources, particularly managerial input, significantly enhances farm productivity across various models, indicating experienced managers utilize technology effectively. Notably, management and human capital innovation drive positive productivity changes in the UK cereal sector for the period 2003–2014, surpassing technological advancements. Efficient farmers leverage experience to benefit from operational scale changes, emphasizing the importance of accumulated knowledge. Hence, policy interventions should recognize these nuances; while promoting vocational training aids technology adoption, it may not spur management innovation. Thus, strategies must balance various aspects to effectively foster innovation in agriculture, considering both technological and managerial advancements for sustained productivity growth. The study advocates for a departure from the 'bigger is better' mentality, proposing educational programs and support services to encourage informed decision-making. This forward-looking approach aims to inform future policies and enhance understanding of the intricate dynamics between agricultural innovation, productivity, and entrepreneurship.

**Keywords:** innovation; farm entrepreneurship; productivity; technical efficiency; farm business management

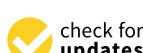



## 1. Introduction

Producing more food, fiber and fuel with fewer inputs, such as land, fertilizers, and water, requires changes to the efficiency with which these inputs are used [1,2]. Productivity improvement not only helps to achieve government goals of "sustainable intensification" (SI) (i.e., producing more with less), as set out, for the United Kingdom (UK), in the Foresight Report on the Future of Food and Farming [3], but it is also beneficial for the individual farm business, as it results in reduced input and resource costs per unit of output, leading to improved margins [4,5]. Furthermore, productivity improvements can also lead to reductions in vulnerability to some risks (e.g., droughts, pests etc.), reductions in escape of damaging chemical inputs to the environment, improvements in product quality, and enhanced social responsibility [6–8].

Productivity increases in a landscape of dynamic entrepreneurship [9]. Entrepreneurs in the agricultural sector play a crucial role in driving innovation, adapting to evolving market demands, and optimizing operational efficiency [10]. Their strategic vision and willingness to embrace new technologies contribute significantly to increased yields, sustainable farming practices, and enhanced overall productivity [11]. Nurturing a supportive environment for entrepreneurship in UK agriculture is essential, as it not only ensures the sector's resilience in the face of challenges but also positions it for long-term success in meeting the growing demands of a dynamic and competitive global market outside of the EU Common Agricultural Policy [11–14].

Various studies have identified, at the farm-business level, a number of drivers of productivity change, for example: structural change (e.g., increasing scale) [15], access to education and information [16], cooperation between farmers and with upstream and downstream actors [17], access to new/improved resources [15], climate change [16,18], and technological diffusion and innovation [16,19], among others. While there are a number of these drivers of productivity change, and the importance of these can vary from farm to farm [20], the Organisation for Economic Co-operation and Development (OECD) concludes that the most important driver of productivity improvement is innovation [21].

## 1.1. Defining Indicators for Innovation

In its very broadest sense, innovation is the generation, diffusion, and exploitation of knowledge [21]. In business/farm management terms, innovation is defined as the introduction of novelty, i.e., some significant change to any of several areas of activity within a business. Traditionally, both the practice and study of innovation has been limited to the adoption of science and technology, but innovation is now perceived in much broader terms, impacting such areas as: technological development (for example, of new products); production techniques; changes to organizational structures and practices; and new marketing operations [21,22]. As with most small firms in the non-agricultural sector (excluding the R&D sector), innovations in farm businesses do not generally originate on the farm itself but are acquired through diffusion of novelty onto the farm from elsewhere.

In EU states and beyond, various governmental and non-governmental studies delve into strategies for fostering innovation in agriculture, shedding light on the pivotal connection between farm-level entrepreneurship and productivity [23]. These initiatives aim to decipher the barriers hindering innovation and emphasize the role of policy in not only incentivizing but also removing obstacles to entrepreneurial activities at the farm level [12]. By correctly aligning policy instruments with the unique challenges faced by agricultural entrepreneurs, the transformative potential to boost productivity, drive technological adoption, and ensure the sustainable growth of the farming sector will be increased [19]. However, policy instruments identified as providing encouragement to innovation are often quite broad in their scope and diffuse In their effects, i.e., directed towards achieving multiple desirable objectives simultaneously, only some of which might be targeted at the type of innovation that leads to productivity improvements. To illustrate, the New Entrants Scheme under the EU Rural Development Regulation (2020) aims to encourage younger farmers into agriculture, on the grounds that they might be more innovative than older farmers [24]. But to what extent and in what ways would this innovation lead to productivity improvement? Also, what is the relative value of this innovation, compared with other sources of innovation, such as increasing levels of training, purchase of new equipment, or increasing the scale of operation, in terms of driving productivity improvement? Would funding be better directed at these things? Before more targeted and informed policy instruments can be designed, the role of different types of innovation in driving productivity change must be better understood.

To assess the efficacy of such innovations in driving productivity gains, it is crucial to be able to quantify, both the innovations (actions) themselves and their productivity change outcomes as well as measure the statistical relationship between them. This raises an important question: what metrics of innovation (as an activity, or input) and productivity

change (as an outcome), specifically for the agriculture sector, in publicly available datasets, are available that would allow these innovation-productivity-change relationships to be illuminated? In agriculture, while it is generally understood that innovation, in some way, drives productivity change, it has proven difficult to quantify this relationship. This difficulty arises not only because of heterogeneity in the population of farm businesses being studied, but also because innovation is a continuous and dynamic process that can occur at any time. Further, innovation can occur in different areas of the business, with each affecting different inputs/resources, and can also be diffuse, i.e., not directed at any one activity (for example, farmer acquisition of new critical decision-making skills). It is simply not the case that innovation only occurs at set times in a business cycle, or when major investments programs are instituted. Innovation occurs when a single new piece of knowledge is brought onto the farm, or a new tool, or a planned small-scale adaptation to management is instituted, leading to change. The piecemeal, random, and multi-scaled nature of innovation presents real challenges for understanding and quantifying the nature of its relationship with productivity. For this reason there have been few attempts to quantify/map this relationship, with some past studies as notable exceptions [25–31]. However, even in the studies cited above, innovation is not studied holistically. Rather, one or more factors of change (e.g., introducing a new piece of technology) that might be deemed to be innovations, are included individually alongside a larger list of possible non-innovation-based drivers of productivity change to study their relative effects.

More often, specific innovations are not themselves identified, but rather proxies for innovation are used, i.e., metrics for activities which might facilitate innovation, or where innovation might conceivably occur. For example, expenditures on consultancy services and training courses are taken to be proxies for innovation because they are assumed to make innovation more likely. However, there is no assurance that in individual cases these proxy activities have led to innovation. Investment in new plant and machinery can be more clearly identified as innovation and it is for this reason that many early studies of the impact of innovation on productivity have been focused on adoption of new technologies [32,33].

Several measures of productivity have been defined in the agricultural economics literature. The measure known as Partial Productivity is defined as the rate of output produced per unit of each input. This measure is obviously too simplistic for use with multi-product firms and so the more holistic measure known as Total Factor Productivity (TFP) was developed. This expresses the ratio between an index of aggregated outputs and an index of aggregated inputs. This articulates the proportion between a consolidated measure of outputs and a consolidated measure of inputs. According to production theory, the factors influencing the rate of output include the technology utilized, the quantity and quality of production factors, and the efficiency with which these factors are utilized in the production process [34]. Consequently, for any farm business, variations in TFP result from the combined impact of changes in efficiency, shifts in the production frontier, and alterations in the scale of production [35].

To study the impact of innovation on productivity in a more holistic way, it is first necessary to identify, from the literature, the broad types of innovation that can occur and select those which are relevant as drivers of productivity change. Within these broad typologies it will be necessary to identify specific innovation metrics that might also be represented in official farm datasets. For these purposes, the universe of innovation has been divided into two broad typologies, i.e., (i) innovation in management (including investment in human capital and entrepreneurial competencies, i.e., training); and (ii) innovation through investment in new technologies. Table 1 below presents the results of this review, identifying each activity found in the literature that might be identified with innovation (i.e., it is innovation directly, or an activity that makes innovation more likely) with its broad typology, as well as providing the source literature.

**Table 1.** Activities identified in the literature as drivers or elements of innovation at farm level.

| Description of Indicator | Literature Source |
|---|---|
| **Management Practices** | |
| Business planning/benchmarking | [31,36–38] |
| Knowledge acquisition use of information sources | [16,27,29,39,40] |
| Use of business management advice | [31,41,42] |
| Machinery sharing | [43–46] |
| Setting goals/targets for business | [47,48] |
| Use of integrated pest management (IPM) | [20,49] |
| Risk management | [41,50–52] |
| Monitoring and evaluation | [27,30,40,53] |
| Record keeping | [40,48,53] |
| Training for IT skills | [16,36] |
| Investment in training programmes (non-IT) | [14,25,28,54–56] |
| Changes to standard operating procedures | [15,47,56,57] |
| New technology innovations | [58–61] |

From the perspective of drawbacks, perhaps the most restrictive barrier to a holistic analysis of the role of innovation in driving farm productivity, is that in the datasets that might be used to derive such metrics, particularly official datasets, there are very few apparent indicators of innovation applicable at the farm level, and consequently, for some dimensions of innovation, no indicators at all. A further complication is that there is no standard metric for measuring productivity change and, consequently, studies in the literature use a range of different metrics of farm performance, such as Total Factor Productivity, output, Net Margin, or simply profitability.

### 1.2. The Purpose and Contributions of the Current Study

This study aims to comprehensively explore the role of innovation, particularly within the context of entrepreneurship in agriculture, and its role in driving transformative changes in farm productivity. The research objectives are: (i) Conduct a thorough literature review—the study aims to map the variables, specifically management actions, that can be reasonably identified as innovation or as factors increasing the likelihood of innovation within agricultural practices. (ii) These identified variables will be classified into either innovation in management or technical change, providing a nuanced understanding of the diverse forms that innovation can take. (iii) Scrutinize data from the Farm Business Survey (FBS) across multiple years, seeking analogues for the indicators pinpointed during the literature search. This approach will use real-world data to validate and contextualize theoretical frameworks. (iv) Identify an appropriate measure of productivity, justifying the chosen metric as a robust indicator of the outcomes resulting from innovative practices. (v) Separately evaluate the impacts of innovation on productivity arising from (a) enhancements in the quality or state of existing farm assets, including human capital, and (b) changes in the scale of farming activity (over time). This differentiation adds granularity to the assessment of innovation's multifaceted effects. (vi) Provide recommendations for future collections of alternative FBS indicators, addressing any deficiencies in the coverage of innovation. This forward-looking approach aims to refine data collection strategies for a more comprehensive understanding of the evolving landscape of agricultural entrepreneurship and innovation.

This study contributes to the literature, primarily, by helping to bridge a gap in our understanding of the relationship between innovation and productivity in agriculture. By mapping variables associated with innovation in farm management and technical changes, and correlating these with productivity measures, we contribute to the exploration of how

innovation manifests and drives efficiency in agricultural practices. The use of longitudinal data from the FBS to validate theoretical constructs provides empirical grounding to the study, enhancing its applicability and relevance for future research.

The remainder of the study is structured as follows. In the Material and Methods section, we present an index of productivity and its components, discussing how these are linked to efficiency in management (innovation) and technical efficiency. We then proceed with a discussion on the use of a panel data model to explore the effects of innovation in management and capital investment on the efficiency and productivity measures obtained by the productivity index and its various components. The Results section presents the findings of both stages, i.e., the 1st stage productivity analysis and the 2nd stage panel data econometric analysis. Both the Materials and Methods section and the Results are supported by Supplementary Materials available for download. The Discussion section elaborates further on the main findings and their contribution to the discussion on innovation and entrepreneurship at a farm level. The manuscript concludes with a review of the key messages and suggestions for future work.

## 2. Material and Methods

### 2.1. Aims and Overview of Methodology

The goal of the study will be achieved in two stages. In the first stage of the analysis, the Malmquist Index (MI) of TFP [62,63] is employed to explore changes in productivity and technical and economic efficiency of farms in the panel over time. The component distance function in the technical change index is then used to identify innovators within the sample, i.e., farms that shift the frontier outwards [34]. In the second stage of the analysis, the factors that enable this innovation will be identified. To facilitate this, the MI of TFP will be decomposed by introducing to the regression analysis the components of change in scale of technology i.e., the product of change in scale efficiency, and change in scale of technology [64–67].

### 2.2. Data Sources

Data used in the modelling exercise are sourced from a representative sample of 60 cereal farms spanning the years 2003–2014. This data was acquired from (FBS) (The Farm Business Survey uses a sample of farms that is representative of the national population of farms in terms of farm type, farm size and regional location (see http://www.farmbusinesssurvey. co.uk and http://www.defra.gov.uk/statistics/foodfarm/farmmanage/fbs/ for details on data collection, methodology, results, among others. Retrieved 20 January 2024)) which provides comprehensive information on the structure and physical and economic performance of farm businesses in England and Wales. The deliberate selection of specialist cereal farms for the sample ensures a relatively homogeneous sample representation in terms of farm system and complexity (Heterogeneity in basic farming system and environmental conditions would add noise to any analysis of the efficiency with which resources are used). The inclusion of 60 cereals farms over a 12-year period results in a panel of 720 observations available for the efficiency analysis. For the assessment of the MI of TFP, this provides 660 observations given that the analysis utilizes data from two consecutive years at a time. The FBS data covers a period of 12 farm accounting periods with the objective of capturing the impact on management efficiency due to the reform of the CAP in 2003 and the subsequent adjustments towards the 2013 CAP reform.

### 2.3. Construction of Variables for the TFP Analysis

The farm output measure is based on market returns from all farm-based enterprises. Non-market sources of revenue (e.g., savings or aid and subsidy payments) are excluded on the grounds that they do not vary in response to changes in production scale, or the quality or quantity of farm inputs used. The exclusion of non-market revenue sources like savings, aid, and subsidies from productivity index calculations is essential for several reasons. Firstly, including these sources would distort productivity measurements as they

are not directly linked to productive farm activities, potentially inflating figures. Secondly, non-market revenue lacks a direct relationship with production inputs, thus inaccurately reflecting input–output dynamics. Thirdly, productivity indices primarily focus on market performance, making non-market sources irrelevant. Moreover, excluding these sources ensures comparability over time and entities, facilitating meaningful productivity assessments. Lastly, maintaining policy neutrality by excluding non-market sources aids in informed policy decisions aimed at improving agricultural efficiency and competitiveness. The composite variable representing production technology, utilized in estimating technical and sub-vector efficiency, as well as the MI of TFP, is constructed from the following components: cultivated area, crop expenses (encompassing fertilizers, crop protection, seeds, and other agricultural costs), and total labor (comprising both paid and unpaid workers). All inputs, expressed in monetary terms (£/ha), have been adjusted to constant price levels based on 2010 prices. (We use using price indices based on 2010 published by the Department for Environment, Food and Rural Affairs (DEFRA) (API—Index of the purchase prices of the means of agricultural production—dataset (2010 = 100)) a, published as "Index of Producer Prices of Agricultural Products, UK (2005 = 100), publication date—18 July 2013". Available online: https://www.gov.uk/government/statistics/agricultural-price-indices [retrieved 22 January 2024]). Specifically, the adjustment involves the use of the following indices: Fertilizers and soil improvement index, seeds index, plant protection products index, farm machinery and installation index, and other costs index.

### 2.4. The Malmquist Index of Total Factor Productivity

To assess the relative performance of farms over time, a dynamic framework is essential. Thus, the time series dimension is employed to analyze shifts in the production frontier, offering insights into technical changes and the progression of individual farms toward the frontier. This approach facilitates the measurement of efficiency changes over time. For this framework, we have chosen an input-oriented Malmquist index, considering that farmers exert more influence over adjusting and efficiently using inputs than expanding output [68]. Specifically, the MI between two periods, $t$ and $t+1$, is defined as the ratio of the distance function for each period in relation to a common technology estimated through Data Envelopment Analysis (DEA), in line with previous studies [66,68]. Therefore, the MI based on an input-distance function is defined as:

$$M_I^t = \frac{D_I^t(x^{t+1}, y^{t+1})}{D_I^t(x^t, y^t)} \tag{1}$$

Equation (1) represents the ratio between the input-distance function for a farm observed at period $t+1$ and period $t$, respectively, when compared to the technology at period $t$. Values of the $M_I < 1$ indicate deterioration (negative changes) in TFP, values of the $M_I > 1$ indicate improvements (positive changes) in TFP, while values of $M_I = 1$ indicate stagnation (no change) in productivity. However, as the selection of period $t$ or $t+1$ as the base year is arbitrary (i.e., the base year can be either period $t$ or period $t+1$), Färe et al. [69] defined the MI of TFP as the geometric mean of the $t$ and $t+1$ $M_I$. Therefore, for each farm the input orientation Malmquist index is expressed as follows:

$$M_I^{t,t+1} = \left[ \frac{D_I^{t+1}(x^{t+1}, y^{t+1})}{D_I^t(x^t, y^t)} \frac{D_I^t(x^{t+1}, y^{t+1})}{D_I^{t+1}(x^t, y^t)} \right]^{1/2} \tag{2}$$

where $M_I^{t,t+1}$ refers to the MI of TFP from period $t$ to period $t+1$; $(x^t, y^t)$ is the farm input-output vector in the $t^{th}$ period; $D_I^t(x^{t+1}, y^{t+1}) = max\left\{ \theta > 0 : \left( \frac{x^{t+1}}{\theta} \right) \in P \right\}$ is the input distance from the observation in the $t+1$ period to the technology frontier of the $t$th period with $P(y^{t+1})$ the input set at the $t+1$ period and $\theta$ is a scalar equal to the efficiency score. The indices are calculated with the use of the nonparametric DEA method (see Supplementary Materials, Section S1 [70,71]) in order to construct a piecewise frontier

that envelopes the data points [35]. The assumption of constant returns to scale (CRS) is employed in estimating the MI of TFP. In the absence of CRS, the measurement of productivity change is not accurate [72]. The primary advantage of the DEA method lies in its ability to avoid misspecification errors and to concurrently assess changes in productivity in scenarios involving multiple outputs and inputs [68]. Additionally, utilizing the DEA method for estimating the MI of TFP simplifies the computation process since DEA does not necessitate information on prices.

In addition, the index in Equation (2) can be decomposed into two components, efficiency change and technological change, as follows:

$$M_I^{t,t+1} = \frac{D_I^{t+1}\left(x^{t+1},\, y^{t+1}\right)}{D_I^t(x^t, y^t)} \times \left[ \frac{D_I^t\left(x^{t+1},\, y^{t+1}\right)}{D_I^{t+1}\left(x^{t+1}, y^{t+1}\right)} \frac{D_I^t\left(x^t,\, y^t\right)}{D_I^{t+1}\left(x^t, y^t\right)} \right]^{1/2} \tag{3}$$

The first component of Equation (3) represents an indicator of relative technical efficiency change, $(\Delta Eff)$, illustrating how much closer or farther a farm moves towards the best practice frontier. Essentially, it gauges the "catch up" effect [69]. The second component serves as an indicator of technical change, $(\Delta Tech)$, revealing the extent of the shift in the frontier. Both components assume values greater than, less than, or equal to unity, signifying improvement, deterioration, or stagnation, respectively, similar to the MI of TFP. In addition, as Färe, Grosskopf, & Lovell [34] and Färe, Grosskopf, Norris, et al. [73] have demonstrated, the index of $\Delta Eff$ is further decomposed into two factors, pure technical efficiency $(\Delta PureEff)$ and scale efficiency change $(\Delta ScaleEff)$.

$$M_I^{t,t+1} = \frac{D_I^{Vt+1}\left(x^{t+1},\, y^{t+1}\right)}{D_I^{Vt}(x^t, y^t)} \times \frac{\frac{D_I^{t+1}\left(x^{t+1}, y^{t+1}\right)}{D_I^{Vt+1}\left(x^{t+1}, y^{t+1}\right)}}{\frac{D_I^t\left(x^t,\, y^t\right)}{D_I^{Vt}(x^t,\, y^t)}} \times \left[ \frac{D_I^t\left(x^{t+1},\, y^{t+1}\right)}{D_I^{t+1}\left(x^{t+1}, y^{t+1}\right)} \frac{D_I^t\left(x^t,\, y^t\right)}{D_I^{t+1}\left(x^t, y^t\right)} \right]^{1/2} \tag{4}$$

where the $D_I^{Vt+1}\left(x^{t+1},\, y^{t+1}\right)$ and $D_I^{Vt}\left(x^t, y^t\right)$ corresponds to distance functions estimated under the variable returns to scale (VRS) assumption. It must also be noted that $\Delta Eff = \Delta PureEff \times \Delta ScaleEff$. The components suggested by Färe, Grosskopf, & Lovell [34] and Färe, Grosskopf, Norris, et al. [73] allow for identification of alterations in the CRS frontier over time $(\Delta Tech)$ and variations in both pure efficiency and scale efficiency corresponding to VRS frontiers from two distinct periods. Additionally, the component distance functions in the technical change index of the MI of TFP identifies the farms responsible for the frontier shift [73]. Specifically:

- if technical change $(\Delta Tech)$ of farm $i$ is greater than 1; and
- the distance function estimates, under CRS, for the farm in the period $t + 1$ relative to estimated technology in period $t$ are also greater than 1; and
- efficiency estimates, under CRS, at time $t + 1$ relative to technology at time $t + 1$ equals 1;
- then that farm has contributed to a shift in the frontier between the two periods. Formally, this is expressed as follows:

$$\Delta Tech^i > 1,\ D_I^t\left(x^{i,\, t+1}, y^{i,t+1}\right) > 1 \text{ and } D_I^{i,t+1}\left(x^{i,\, t+1}, y^{i,t+1}\right) = 1 \tag{5}$$

Kneip et al., [74], Simar & Wilson [64,65], and Wheelock & Wilson [67] introduced an additional decomposition of the MI of TFP to assess changes in technology through alterations in the VRS estimate. Specifically, if a farm's position remains constant in the input-output space during periods $t$ and $t + 1$, with the only change occurring in the VRS estimate of technology, then $(\Delta Tech)$ in Equation (4) will be equal to unity, signifying no change in technology. Therefore, for a change in technology to be indicated, the CRS estimate of technology must change. Hence, Kneip et al. [74], Simar & Wilson [64,65] proposed the following decomposition, based on the assumptions of Kneip et al. [74] that the VRS estimator is always consistent:

$$M_I^{t,t+1} = \frac{D_I^{Vt+1}\left(x^{t+1}, y^{t+1}\right)}{D_I^{Vt}(x^t, y^t)} \times \frac{\frac{D_I^{t+1}\left(x^{t+1}, y^{t+1}\right)}{D_I^{Vt+1}\left(x^{t+1}, y^{t+1}\right)}}{\frac{D_I^t\left(x^t, y^t\right)}{D_I^{Vt}(x^t, y^t)}} \times \left[\frac{D_I^{Vt+1}\left(x^t, y^t\right)}{D_I^{Vt+1}\left(x^{t+1}, y^{t+1}\right)} \frac{D_I^{Vt}\left(x^t, y^t\right)}{D_I^{Vt}\left(x^{t+1}, y^{t+1}\right)}\right]^{1/2}$$
$$\times \left[\frac{\frac{D_I^{t+1}\left(x^t, y^t\right)}{D_I^{Vt+1}\left(x^t, y^t\right)}}{\frac{D_I^{t+1}\left(x^{t+1}, y^{t+1}\right)}{D_I^{Vt+1}\left(x^{t+1}, y^{t+1}\right)}} \frac{\frac{D_I^t\left(x^t, y^t\right)}{D_I^{Vt}(x^t, y^t)}}{\frac{D_I^t\left(x^{t+1}, y^{t+1}\right)}{D_I^{Vt}\left(x^{t+1}, y^{t+1}\right)}}\right]^{1/2}$$

(6)

where the first two components indicate $\Delta PureEff$ and $\Delta ScaleEff$ and the $\Delta Tech$ is decomposed into pure technical ($\Delta PureTech$) and scale technical change ($\Delta ScaleTech$). Also, $\Delta Tech = \Delta PureTech \times \Delta ScaleTech$. The index of pure technical change serves as the geometric mean of these two ratios, reflecting shifts in the VRS frontier between the two periods. Values of $\Delta PureTech$ exceeding unity indicate an expansion in pure technology, values below unity suggest a deterioration, and values equal to unity signify stagnation in pure technology. Information obtained from the scale technology change index is utilized to characterize alterations in returns to scale of the VRS frontier between two time periods. Values of ($\Delta ScaleTech$) greater than unity imply that farms operate either below or above the optimal scale, values below unity suggest movement CRS, and when it equals unity, there are no changes in the shape of technology.

### 2.5. Panel Data Econometric Models

A set of econometric models are estimated using the random effects and the Feasible Generalised Least Squares (FGLS) procedures for panel data. Financial and management characteristics data is derived from the FBS and used in a second stage regression to explore the effects of innovation in management and investment on the efficiency and productivity measures obtained by the MI of TFP and its various components. The decomposition of the MI of TFP to its various components and how these are used to capture innovation at a farm level are illustrated in Figure 1.

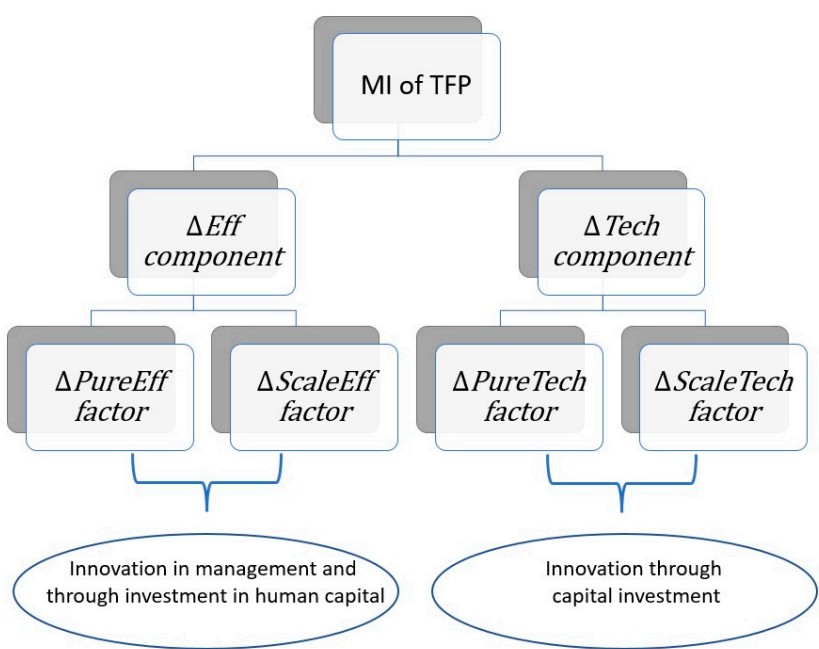

**Figure 1.** Decomposing the MI of TFP to capture innovation at a farm level.

The efficiency change component ($\Delta Eff$) of the MI of TFP and its two factors, pure technical efficiency ($\Delta PureEff$) and scale efficiency change ($\Delta ScaleEff$), are regressed to a set of variables in order to explore further how technically efficient managers capture the productivity gains observed in the various periods. Hence, conclusions in regard to

innovation in management and through investment in human capital could be derived. In particular, $\Delta Eff$ and its two factors were regressed against the form of business (Sole trader, Partnership, and Farming Company), the age of the farmer, the level of education (basic education, i.e., school only, and further education), a dummy variable indicating paid or not managerial input, and the size of the farm (based on the FBS classification of size). In addition, a dummy variable was used to indicate those farms that are owner-occupied or tenanted, and an index to define the level of specialization for each farm in producing arable crops was designed. The index of specialization considered the output derived from arable enterprises and the output derived from livestock enterprises. A farm will receive an index of 1 when all its output is derived from arable enterprises and any other number less than that will identify the percentage of other enterprises contributing to the total output of the farm business. Hence, three levels of specialization were defined for all farms through the periods under consideration (Level 1: 0.7–1, Level 2: 0.5–0.69, and Level 3: 0–0.49).

Descriptive statistics for the variables used for the innovation in management and human capital panel data models are available in Table 2. Key insights show that the average Efficiency Change is slightly above 1 (Mean = 1.04, SD = 0.25), indicating a general improvement in efficiency for the cereals farm sample over the period studied. The Pure Efficiency Change, which represents efficiency improvements excluding scale and mix effects, is close to 1 (Mean = 1.01, SD = 0.14), suggesting modest gains. The Scale-mix Efficiency Change, reflecting changes due to scale and mix of outputs, is also slightly above 1 (Mean = 1.03, SD = 0.18).

Farm structure (see Table 2) was predominantly sole traders (47%) or partnerships (45%). Limited companies comprise a smaller portion (8%). This distribution suggests a dominance of traditional and family-run farm businesses. The education and management characteristics of farmers show a significant majority of the farmers achieving some higher education qualification (64%), with fewer having only basic education (17%) or A-level qualifications (19%). However, most farms (95%) operate without paid managerial input, highlighting the reliance on the farmers' own expertise. The distribution between large (45%) and medium-sized (44%) farms is fairly even, with small farms making up only a minor proportion (11%). The majority of farms are owner-occupied (65%) rather than tenanted (35%). Also, a majority of farms (89%) have more than 70% of their output in crops, indicating a strong focus on crop production in the sample, as would be expected. Data on farmers' age suggest an apparent aging of the farmer population over the survey period, with the average age increasing from 53 in 2003/2004 to 62 in 2013/2014. Overall, descriptive data suggest that while there have been slight improvements in farm efficiency, these are more attributable to scale and mix changes rather than pure efficiency.

For quality assurance purposes, a series of specification tests were performed on the panel data models (Hausman-type tests). In addition, a series of diagnostic checks were used regarding serial correlation, heteroscedasticity, and cross-sectional dependence. Hence, the model specified for the $\Delta Eff$ and its two factors ($\Delta PureEff$, $\Delta ScaleEff$) was a random effects model. Moreover, since heteroscedasticity has been detected in the case of innovation in management and human capital ($\Delta Eff$, $\Delta PureEff$, $\Delta ScaleEff$), a robust covariance matrix has been used to account for it.

**Table 2.** Descriptive statistics of the 2nd stage regression variables to link the Efficiency components of the MI of TFP (FBS data, 11-year period) with innovation in management change at a farm level.

| Dependent Variables | | | Mean | SD |
|---|---|---|---|---|
| Efficiency Change ($\Delta Eff$) | | | 1.04 | 0.25 |
| Pure Efficiency Change ($\Delta PureEff$) | | | 1.01 | 0.14 |
| Scale-mix Efficiency Change ($\Delta ScaleEff$) | | | 1.03 | 0.18 |
| List of Independent Variables % of N = 660 | | | | |
| Sole trader | 47% | Paid Managerial Input | | 5% |
| Company | 8% | No managerial input | | 95% |
| Partnership | 45% | Large size farms [1] | | 45% |
| Holder Manager | 87% | Medium size farms | | 44% |
| Holder not Manager | 5% | Small size farms | | 11% |
| Limited Company | 8% | Tenanted farms (majority of tenanted land) | | 35% |
| Basic Education only | 17% | Owned farms | | 65% |
| A-Level or Equivalent | 19% | Crop output less than 50% of total | | 4% |
| Higher education | 64% | Crop output more than 50% and less than 70% | | 7% |
| | | Crop output more than 70% | | 89% |
| Farmers Age | | | Mean | SD |
| 2003/2004 | | | 53 | 9.6 |
| 2013/2014 | | | 62 | 9.4 |

Farm Businesses are classified by size according to the Standard Labor Requirements (SLRs). SLRs are calculated for different livestock and crop types and provide an estimate of the total amount of standard labor used on the farm. This leads to the classification of farms by number of full-time equivalent (FTE) workers as follows: Small 1 < 2 FTE, Medium 2 < 3 FTE, Large ≥ 3 FTE. More information is available here: https://assets.publishing.service.gov.uk/media/641073c8e90e076cd09acda9/fbs-uk-farmclassification-2014-14mar23.pdf. (accessed on 17 March 2023).

## 3. Results

### 3.1. The MI of TFP and Its Components

A detailed statistical analysis of the MI of TFP for the period 2003 to 2014 is detailed in Supplementary Materials, Table S1.

The most important shifts in productivity are identified in the periods 2008–2009 (MI = 1.248) and 2011–2012 (MI = 1.27). The lowest average level of productivity is observed in the period between 2009 and 2010 (MI = 0.791). Variation in the average value of the MI and its components (efficiency and technical change component) is shown in Figure 2. We can observe that the significant regressions or advancements of the MI are mainly caused by the technical change component rather than the efficiency change component, which is approaching unity in most periods. A significant deterioration of the technical change component is observed between the periods 2003/2004 and 2006/2007. With some fluctuation, MI is constantly under improvement after the 2009/2010 period as all scores are above unity. The product of efficiency and technical change should by definition be equal to the MI in each period.

Table 3 provides additional information regarding TFP changes for each farm size group over time. To assess potential statistically significant distinctions in productivity changes among farm size groups, the Kruskall–Wallis test (one-way analysis of variance by ranks) was employed. The null hypothesis, asserting that sub-samples are drawn from the same distribution, could not be rejected for any period. This suggests an absence of significant differences in productivity change among different farm sizes throughout the study period. All farm size categories exhibit an MI value below unity. Moreover, the average MI for the 11-year period for large, medium, and small farms is 0.98, 0.98, and 0.97, respectively, indicating a slight decline in productivity over the period. The corresponding geometric means for efficiency change per farm size group for the same period are 1.01 (small size), 1.02 (medium size), and 1.03 (large size). In terms of the technical change

component, the average ΔTech for large, medium, and small farms is 0.96, 0.98, and 0.98, respectively. Consequently, the geometric means for ΔTech and the ΔEff suggest that any improvement in the MI of TFP over the period is primarily attributed to innovation in management and investment in human capital rather than innovation through investments in new technology.

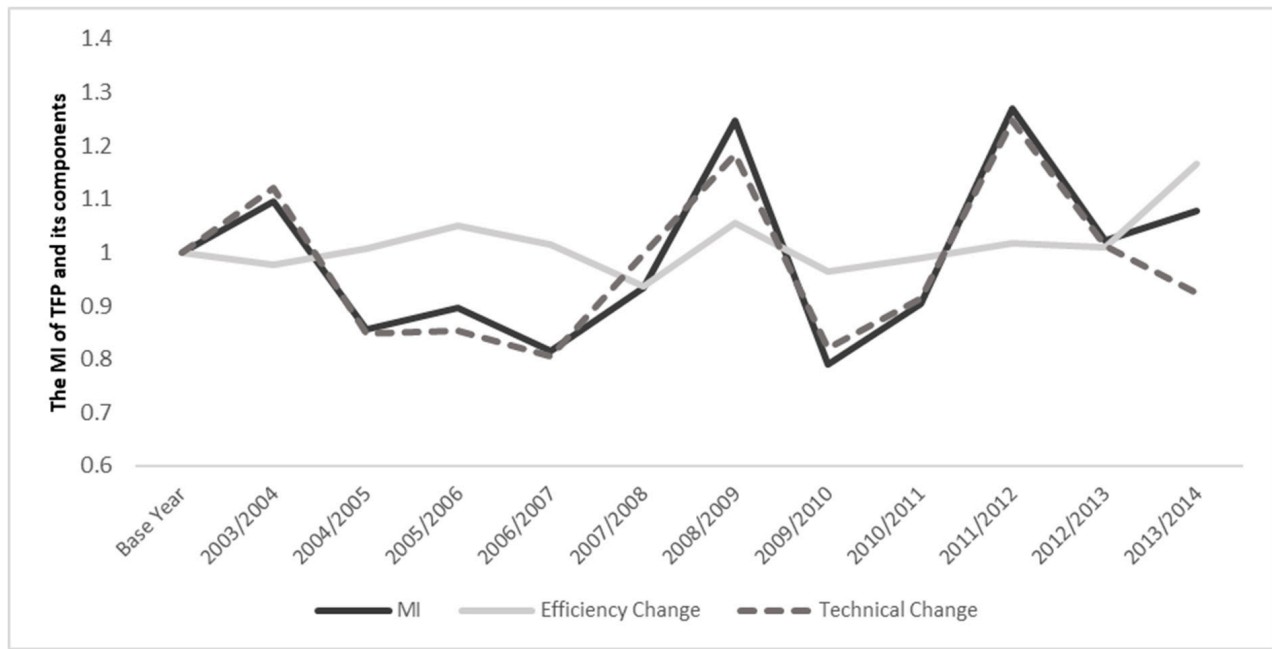

**Figure 2.** Total factor productivity, efficiency, and technical change for the 11-year period.

**Table 3.** The MI of TFP (Malmquist Index) per year and per farm size.

| Farm Size | 2003/2004 | | 2004/2005 | | 2005/2006 | | 2006/2007 | |
|---|---|---|---|---|---|---|---|---|
| | **Mean** | **SD** | **Mean** | **SD** | **Mean** | **SD** | **Mean** | **SD** |
| Large | 1.11 | 0.2 | 0.84 | 0.15 | 0.87 | 0.13 | 0.83 | 0.13 |
| Medium | 1.11 | 0.25 | 0.9 | 0.23 | 0.94 | 0.25 | 0.82 | 0.21 |
| Small | 1.12 | 0.16 | 0.86 | 0.14 | 0.91 | 0.21 | 0.88 | 0.29 |
| Farm Size | 2007/2008 | | 2008/2009 | | 2009/2010 | | 2010/2011 | |
| | Mean | SD | Mean | SD | Mean | SD | Mean | SD |
| Large | 0.9 | 0.18 | 1.43 | 0.3 | 0.84 | 0.45 | 0.95 | 0.23 |
| Medium | 0.95 | 0.24 | 1.29 | 0.34 | 0.82 | 0.19 | 0.91 | 0.22 |
| Small | 1.03 | 0.35 | 1.25 | 0.39 | 0.81 | 0.2 | 0.94 | 0.26 |
| Farm Size | 2011/2012 | | 2012/2013 | | 2013/2014 | | | |
| | Mean | SD | Mean | SD | Mean | SD | | |
| Large | 1.27 | 0.37 | 1.06 | 0.3 | 1.13 | 0.22 | | |
| Medium | 1.4 | 0.41 | 1.05 | 0.25 | 1.11 | 0.25 | | |
| Small | 1.25 | 0.29 | 1.05 | 0.26 | 1.1 | 0.32 | | |

Note: Since the Malmquist index is multiplicative, these averages are also multiplicative (i.e., geometric means).

### 3.2. Test for Innovators in the Sample

During the periods 2005/2006, 2006/2007, and 2009/2010 no farm caused any outward shift to the frontier since technical change was less than unity for all farms. In total, 25 farms have been identified as responsible for the outward frontier shift in the remaining accounting periods (in particular, farms 1, 2, 9, 13, 14, 18, 21, 26, 30, 31, 32, 33, 34, 35, 36, 38, 39, 42, 43, 45, 46, 51, 55, 59 and 60). Based on the principle outlined in Section 2.4, these farms can be identified as the "innovators" in the sample.

### 3.3. Decomposition of the Efficiency Change Index into Pure Efficiency Change and Scale Efficiency Change

The efficiency change index can be further decomposed into pure efficiency and scale efficiency change, thereby allowing for the isolation of the impact of changes to farm scale on efficiency change. Table 4 reports the distribution of pure and scale efficiency estimates over the review period (estimates of pure and scale efficiency per farm are presented in Tables S2 and S3 in the Supplementary Materials).

As Table 4 shows, for 2009/2010, the scale efficiency index has experienced enhancement for over 71% of the sample farms. In contrast, the pure efficiency index deteriorates for 51% of the farms in the sample. Figure 3 illustrates that scale efficiency undergoes a decline immediately following the 2008/2009 period, possibly influenced by a loss of confidence or a tighter money supply following the financial crisis, but subsequently recovers, forming an upward trend. Furthermore, the improvement in aggregate efficiency for 2008/2009 is primarily attributed to enhancements in pure efficiency and subsequently closely follows the pure efficiency trend. In summary, the main contributing factor to the improvement in the efficiency change index for the 2010/2011 period is pure efficiency.

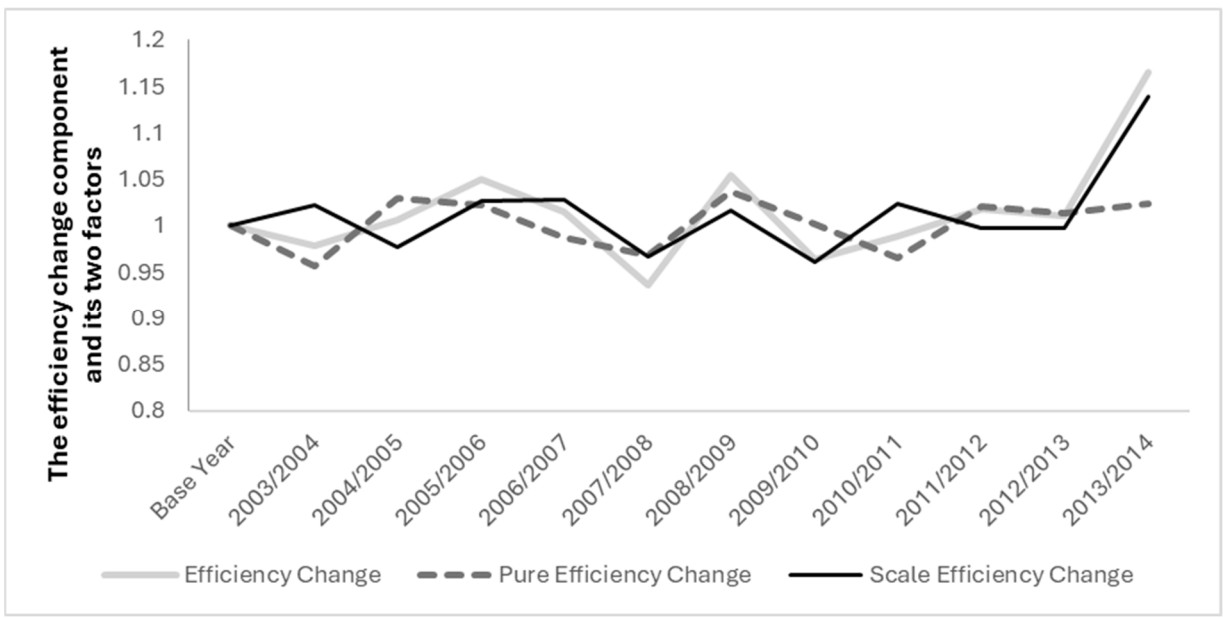

**Figure 3.** The change in efficiency component of the MI of TFP and its two factors, pure efficiency change and scale change over the 11-year period.

**Table 4.** Distribution of the pure and scale factors of the efficiency change component ($\Delta Eff$) over the 11-year period.

| Distribution | 2003/2004 Pure No. of Farms | 2003/2004 Scale No. of Farms | 2004/2005 Pure No. of Farms | 2004/2005 Scale No. of Farms | 2005/2006 Pure No. of Farms | 2005/2006 Scale No. of Farms | 2006/2007 Pure No. of Farms | 2006/2007 Scale No. of Farms | 2007/2008 Pure No. of Farms | 2007/2008 Scale No. of Farms | 2008/2009 Pure No. of Farms | 2008/2009 Scale No. of Farms |
|---|---|---|---|---|---|---|---|---|---|---|---|---|
| <0.6 | 0 | 0 | 0 | 1 | 0 | 0 | 0 | 0 | 0 | 0 | 0 | 0 |
| $0.6 \leq Eff < 0.8$ | 7 | 2 | 0 | 3 | 2 | 2 | 3 | 1 | 7 | 8 | 2 | 4 |
| $0.8 \leq Eff < 1$ | 13 | 18 | 13 | 27 | 14 | 23 | 17 | 22 | 16 | 19 | 14 | 13 |
| $Eff = 1$ | 25 | 5 | 24 | 3 | 24 | 4 | 25 | 4 | 22 | 6 | 24 | 8 |
| $1 < Eff < 1.2$ | 12 | 27 | 18 | 24 | 15 | 26 | 11 | 26 | 11 | 19 | 11 | 29 |
| $1.2 \leq Eff < 1.4$ | 2 | 6 | 3 | 2 | 4 | 3 | 4 | 3 | 3 | 7 | 7 | 5 |
| $Eff > 1.4$ | 1 | 2 | 2 | 0 | 1 | 1 | 0 | 2 | 1 | 0 | 1 | 1 |
| Improvement | 25% | 58% | 38% | 43% | 33% | 50% | 25% | 52% | 25% | 43% | 32% | 58% |
| Deterioration | 33% | 33% | 22% | 52% | 27% | 42% | 33% | 38% | 38% | 45% | 27% | 28% |
| Geometric Mean | 0.95 | 1.02 | 1.03 | 0.98 | 1.02 | 1.03 | 0.99 | 1.03 | 0.97 | 0.97 | 1.04 | 1.02 |

| Distribution | 2009/2010 Pure No. of Farms | 2009/2010 Scale No. of Farms | 2010/2011 Pure No. of Farms | 2010/2011 Scale No. of Farms | 2011/2012 Pure No. of Farms | 2011/2012 Scale No. of Farms | 2012/2013 Pure No. of Farms | 2012/2013 Scale No. of Farms | 2013/2014 Pure No. of Farms | 2013/2014 Scale No. of Farms |
|---|---|---|---|---|---|---|---|---|---|---|
| <0.6 | 1 | 0 | 0 | 0 | 0 | 0 | 0 | 1 | 0 | 0 |
| $0.6 \leq Eff < 0.8$ | 3 | 5 | 6 | 5 | 2 | 9 | 3 | 3 | 5 | 1 |
| $0.8 \leq Eff < 1$ | 14 | 29 | 18 | 14 | 14 | 13 | 14 | 24 | 8 | 7 |
| $Eff = 1$ | 22 | 7 | 24 | 8 | 24 | 6 | 23 | 4 | 23 | 4 |
| $1 < Eff < 1.2$ | 16 | 15 | 10 | 25 | 14 | 26 | 15 | 22 | 15 | 25 |
| $1.2 \leq Eff < 1.4$ | 2 | 3 | 1 | 5 | 5 | 5 | 4 | 4 | 8 | 15 |
| $Eff > 1.4$ | 0 | 0 | 1 | 2 | 1 | 0 | 0 | 2 | 1 | 5 |
| Improvement | 30% | 30% | 20% | 53% | 33% | 52% | 32% | 47% | 40% | 75% |
| Deterioration | 30% | 57% | 40% | 32% | 27% | 37% | 28% | 47% | 22% | 13% |
| Geometric Mean | 1.00 | 0.96 | 0.96 | 1.02 | 1.02 | 0.99 | 1.01 | 0.99 | 1.02 | 1.14 |

### 3.4. The Determinants of Innovation in Management and Innovation through Human Capital

Table 5 presents the results from the three panel data regression models accounting for random effects using the DEA estimates of the change in aggregate efficiency and its two components as dependent variables. The purpose of each regression model is to identify the parameters which have a significant impact as determinants of innovation in management and innovation through investment in human capital (the proxy for this is the presence of paid managerial input). The following model has been estimated using the efficiency change component and the pure efficiency and scale efficiency change sub-components respectively as dependent variables:

$$y_{it} = \beta_0 + \beta_1 SolTr_{it} + \beta_2 Comp_{it} + \beta_3 FarmAge_{it} + \beta_4 HoldMan_{it} +$$
$$\beta_5 BasicEdu_{it} + \beta_6 Alevel_{it} + \beta_7 PaidMan_{it} + \beta_8 Medium_{it} + \beta_9 Small_{it} +$$
$$\beta_{10} Tenant_{it} + \beta_{11} Spec(<50)_{it} + \beta_{12} Spec(>50,<70)_{it} + \alpha_i + u_{it}$$

where $\alpha_i \sim iid(0, \sigma_a^2)$ and $u_{it} \sim (0, \sigma_u^2)$. When the $\Delta Eff$ component is considered as the dependent variable for the model (MD1), estimation results reveal a positive and statistically significant effect when the form of business is a company, compared to a partnership or sole trader ($\beta_2 = 0.059$, $p$-value $< 0.05$). The magnitude of the effect is reduced ($\beta_2 = 0.029$) when the $\Delta PureEff$ factor is considered as the dependent variable in the model (MD2) but it remains statistically significant at $\alpha = 0.05$. However, when the $\Delta ScaleEff$ is considered as the dependent variable of the model (MD3), the effect, although positive, is no longer statistically significant ($p$-value $> 0.05$. For both MD1 and MD3, the effect of an increase in the age of the farmer by one unit is positive across time and across individual farmers; however, it is small in magnitude ($\beta_{3MD1} = 0.001$ and $\beta_{3MD3} = 0.001$, $p$-value $< 0.05$ and $p$-value $< 0.10$, respectively).

In regards to the farmer being both the owner and the manager of the farm, Table 5 shows for all three models that the effect is positive ($\beta_{4MD1} = 0.070$, $\beta_{4MD2} = 0.030$, $\beta_{4MD3} = 0.038$) and significant ($p < 0.05$). Basic education (i.e., school only), has also a positive and significant effect ($\beta_5 = 0.030$, $p$-value $< 0.01$) for MD1 and for MD2 ($\beta_5 = 0.011$, $p$-value $< 0.10$) when compared with higher levels of education (i.e., degree, college, and post-graduate studies). Interestingly, the effect of A-level or equivalent studies is negative for both the MD1 and MD3 models ($\beta_6 = -0.035$, $\beta_6 = -0.024$, $p$-value $< 0.05$). In terms of innovation through investment in human resources, the paid managerial input has a significant and positive effect in all three models and is the strongest parameter in terms of magnitude of the coefficient ($\beta_{7MD1} = 0.132$), indicating that farms with trained and experienced farm managers make better use of the existing technologies and are able to retain this over subsequent periods.

With respect to farm size, results from Table 5 suggest that medium and small farms are less able to achieve a positive effect on all three DVs than large farms, but only medium size is significant at the 5% level ($\beta_{8MD1} = -0.026$, $p$-value $< 0.05$). This indicates that medium size farms drive a smaller change in $\Delta Eff$ than large farms and their average efficiency change across time is 0.026 less than of the average of large farms. Moreover, the results indicate that tenanted farms, on average, across time and across individuals, drive a higher level of efficiency change when compared with owned farms for all three models ($\beta_{10MD1} = 0.026$, $\beta_{10MD2} = 0.014$, $\beta_{10MD3} = 0.013$, $p$-value$_{MD1,MD2} < 0.05$, $p$-value$_{MD3} < 0.10$). In addition, the estimation results regarding level of specialization (i.e., the business output derived from crop enterprises or other enterprises) indicate that the more diverse the farm business output is (less than 50% crop output), the higher the average efficiency change across time and individual farm business when compared to farm businesses where the percentage of crop output is more than 70% ($\beta_{11MD1} = 0.090$, $p$-value $< 0.05$). In contrast, a negative average change of efficiency is estimated by MD1 for the level of specialization between 50% and 70%; however, this is only statistically significant for MD1 ($p$-value $< 0.10$).

**Table 5.** Panel data random effects regression results of the Δ*Eff* component and its two factors Δ*PureEff* and Δ*ScaleEff*.

| Independent Variables | Dependent Variable Efficiency Change Component | | Dependent Variable Pure Efficiency Change | | Dependent Variable Scale Efficiency Change | |
|---|---|---|---|---|---|---|
| | Estimated Coefficient | Standard Error | Estimated Coefficient | Standard Error | Estimated Coefficient | Standard Error |
| Intercept | 0.920 *** | 0.039 | 0.972 *** | 0.023 | 0.948 *** | 0.031 |
| Sole Trader | −0.011 | 0.009 | −0.001 | 0.005 | −0.008 | 0.007 |
| Company | 0.059 ** | 0.023 | 0.029 ** | 0.014 | 0.026 | 0.019 |
| Farmer's Age | 0.001 ** | 0.000 | 0.000 | 0.000 | 0.001 * | 0.000 |
| Holder Manager | 0.070 *** | 0.019 | 0.030 *** | 0.011 | 0.038 ** | 0.015 |
| Basic Education | 0.030 *** | 0.011 | 0.011 * | 0.006 | 0.013 | 0.009 |
| A-Level or Equivalent | −0.035 ** | 0.015 | −0.012 | 0.009 | −0.024 ** | 0.012 |
| Paid Managerial Input | 0.132 *** | 0.023 | 0.028 ** | 0.014 | 0.093 *** | 0.018 |
| Medium Size farm | −0.026 ** | 0.011 | −0.020 *** | 0.006 | −0.007 | 0.009 |
| Small Size farm | −0.010 | 0.018 | −0.016 | 0.011 | 0.004 | 0.014 |
| Tenanted farm | 0.026 ** | 0.010 | 0.014 ** | 0.006 | 0.013 * | 0.008 |
| Crop output less than 50% | 0.090 ** | 0.033 | 0.000 | 0.020 | 0.088 *** | 0.026 |
| Crop output more than 50% and less than 70% | −0.045 * | 0.025 | −0.016 | 0.015 | −0.030 | 0.020 |
| | Balanced data: n = 60, T = 11, N = 660, $R^2$ = 0.09, F-statistic = 5.348 *p*-value < 0.001 | | Balanced data: n = 60, T = 11, N = 660, $R^2$ = 0.05, F-statistic = 2.898 *p*-value < 0.001 | | Balanced data: n = 60, T = 11, N = 660, $R^2$ = 0.06, F-statistic = 3.661 *p*-value < 0.001 | |

Significance codes: '***' < 0.01 '**' < 0.05 '*' < 0.1.

## 4. Discussion

### 4.1. What Has Been Driving Productivity Change?

The decomposition of the MI of TFP has permitted further exploration of the drivers of productivity change in the specialist cereals farm sector over the study period [73] and, in particular, highlighted the impacts of innovation on improvements in management efficiency and technological progress [75,76]. Significant variation, i.e., periods of improvement and regression, in the MI of TFP is observed over the 11-year period. The fact that productivity change is, in some periods, negative, highlights the warning given by Glendining et al. [77] that maintaining productivity per unit area is an important requirement for the future sustainability of arable farming systems. Of great relevance to policy makers is the finding that, over the study period, it is the innovation in management and in human capital that drives positive productivity changes in the UK cereal sector, rather than technological innovation.

### 4.2. Managerial and Entrepreneurial Efficiency

The geometric mean of the Δ*Eff* component for the 11-year period is above unity. This strongly suggests that cereals farmers have been successful in adopting innovations sufficient to improve management and enhance human capital, so that they can solve problems and make relatively efficient resource allocations at the farm level [78]. Focusing on the two sub-components of the Δ*Eff* index, it is noted that it is the scale efficiency (Δ*ScaleEff*) component that is actually driving positive productivity change. This observation confirms the conclusions of [6], that over the study period, management efficiency gains have been driven largely by increasing the scale of operations, rather than by improving the quality of management.

According to Färe, Grosskopf, Norris, et al. [73], analysis of the different components of the MI of TFP allows for the identification of the specific decision-making units driving positive shifts in the efficiency frontier and also allows for the identification (and description) of the best performing farms (in terms of productivity change) in the sample. Positive productivity change occurs because innovation occurs on farms, leading to more efficient use of resources. Therefore, the best performing farms (in terms of productivity change) are, by definition, the most innovative. In theory, by describing these 'leading' farms using key variables, it would be possible to use them for benchmarking purposes, i.e., identifying from these farms those changes to management practices (i.e., innovations) that would improve the productivity of 'lagging' farms. In practice, however, benchmarking attempts break down, because farms that innovate, do not do so consistently, i.e., in some years they contribute to a positive shift in the productivity frontier and in others they pull it back. This is due to the fact that while a farm may innovate, perhaps through investment in training in year 1, it will enjoy productivity improvements in years 2 and 3, but if no further innovations are made, lagging farmers catch up.

### 4.3. A Word on Economies of Scale

A number of studies in the academic literature comment on the relationship between resource use efficiency and growth in farm size (see e.g., [79,80]). The conclusion of these studies is that greater productivity gains are observed in farms expanding the scale of their operations. Realizing economies of scale at a farm level is therefore considered as an important means to improve the productivity of agricultural systems [81]. In designing its own agricultural policies post-Brexit, UK policy makers will be confronted with the challenge of ongoing market-driven consolidation in the agricultural sector and will have to take decisions on whether to allow this process to continue [82]. Grant [82] has shown that increasing scale does appear to increase management efficiency and lead to productivity gains. The question is, how long can this continue? The UK already has among the largest average farm sizes in Europe. Is there an optimal farm size, in terms of management efficiency, in a UK context, beyond which regression occurs? Whether this is the case or not, it is clear from this analysis that UK farmers have had a kind of monomania in looking to economies of scale as a means to increasing productivity, at the expense of alternatives. It should be clear to policy makers, therefore, that there are unrealized opportunities to further improve farm productivity that could be found through a policy focus on programs designed to improve the quality of management. A good step in this direction would be government incentives to increase the use of decision-support tools for agriculture at the farm level, as these have been demonstrated to significantly contribute to the improvement of productivity (agricultural outputs) and environmental outputs [83,84]. Although a plethora of these tools are available, their rate of uptake by UK farmers to this point is low [85].

### 4.4. Management and Technology as Drivers of Innovation

The $\Delta Eff$ factor and its two sub-components were used in a second stage regression analysis to explore the role that farm and farmer characteristics may have on the level of innovation in management and capital investment in human capital. Although findings in the productivity and technical efficiency (at the farm level) literature indicate that productivity, and hence efficiency, of farmers decreases with age [86,87] we found a positive relationship between age and $\Delta Eff$. This positive effect is mainly derived from the positive and statistically significant scale efficiency change. The latter indicates that in terms of scale efficiency, the technically efficient farmer in the sample has the experience and knowledge accumulated over the years to capture the productivity gains associated with changing the scale of operations. Nowak et al. [88] and Gadanakis et al. [83] both also report that length of management experience is positively correlated with improvements in productivity and technical efficiency. Furthermore, the positive relationship of basic education with management efficiency, in combination with the findings regarding age, suggests that

experience and knowledge accumulated over the years can be a substitute for higher levels of formal education [88,89].

However, these trends are not replicated when we consider $\Delta Tech$, where higher levels of education have a positive influence in improving the technology and positively shifting the technical efficiency frontier [86]. These observations mean that policy interventions designed to encourage the adoption of innovation in the agricultural sector must be nuanced enough to capture some of these apparent contradictions. For example, a policy instrument to encourage innovation based solely on vocational training/knowledge transfer may yield desirable results in terms of adoption of new technologies but have very little impact on innovation in management.

An area of particular interest is the impact of investment in innovation in human capital. In terms of efficiency change of MI of TFP, as indicated by the sign of the coefficient for the paid managerial input and for the case when the farm holder is also the manager, we expect further improvements in productivity and technical efficiency. The same is concluded for both the pure and scale efficiency change factors. However, these findings require further investigation in order to explore further the management style and the decision-making process at farm level. According to Pollak [90] and Gallacher et al. [91], professional management might be more conducive to productivity gains compared to management provided by a family member. This is probably because professional managers tend to be better educated and exhibit greater managerial ability, including greater attention to detail, than their owner-occupier counterparts.

Moreover, Byma [26] also found that measured efficiency is influenced by managerial ability and that older and more educated farmers show higher efficiency, as do larger farms which is in line with the findings presented here. However, Byma [26] also suggests that more work is required in understanding the determinants of managerial ability. In addition, it is also important to consider the fact that farms operating under a company status are more likely to observe an improvement in technical efficiency; presumably because there is greater pressure to make profits. Once more, the management style and the decision-making process require further understanding to allow these outcomes to be translated into specific strategies and recommendations for policy makers.

Higher levels of specialization in terms of farm activities were expected to be linked to positive improvement in $\Delta Eff$. However, the results show an inverse relationship, i.e., farms with a percentage of crop output less than 50% are more likely to realize a positive efficiency change. This is mainly due to the scale-mix efficiency change. Since the output considered in the DEA linear programming problem is the farm business output (excluding subsidies) it can be said that during times of high input price inflation, mixed farms, which rely less on imported inputs, have lower costs and seem relatively more efficient in use of inputs. Nevertheless, for the purposes of this study, further investigation is required to validate these findings. This validation might also benefit from the availability of different indicators of farm specialization as the FBS indicator currently available is rather crude, with limited coverage.

### 4.5. A Word on Entrepreneurial Competencies in Agriculture

Entrepreneurship plays a crucial role in the agricultural sector, linking to farm management, human capital innovation, and productivity in several ways [14]. When business development and innovation is linked to farm management, it is necessary to consider the functions of planning and control of farm systems, i.e., explore the dynamics in decision making, resource allocation, and risk management. Entrepreneurs in agriculture make critical decisions regarding crop selection, land use, resource allocation, and technology adoption [12–14]. Effective farm management involves strategic planning, risk assessment, and efficient utilization of resources to maximize productivity. This involves optimizing the use of land, water, fertilizers, and other inputs to ensure sustainable and profitable farming operations. Furthermore, agriculture is inherently risky due to factors like weather conditions, market fluctuations, and pest outbreaks. Therefore, entrepreneurial skills are

essential for managing and mitigating these risks through diversification, insurance, and other risk management strategies.

Human capital innovation is a driver of technology adoption, diversification, training, and skill development. Farmers acting as entrepreneurs lead in the adoption of innovative technologies, including the use of precision farming techniques, data analytics, and other advanced tools to improve productivity and reduce resource wastage [16,19]. Moreover, entrepreneurial farmers invest in continuous training and skills development. This enhances the human capital in agriculture by improving the efficiency and effectiveness of farm operations. In addition to the adoption of new technologies, entrepreneurial farmers are also more likely to explore new crops, farming techniques, and value-added products. Thus, linking this back to training and skills development in the agricultural sector since these activities require a well-trained and adaptable workforce capable of embracing and implementing new ideas and technologies.

Entrepreneurship in agriculture drives efficiency improvements through the constant search for ways to increase productivity and minimize waste throughout farm operations, encompassing streamlined supply chains, optimized logistics, and judicious use of inputs. Entrepreneurs in this sector also maintain a market-oriented approach, producing goods in line with consumer demand to enhance competitiveness and boost sales, ultimately leading to increased profitability [4,5]. In summary, agricultural entrepreneurs foster innovation by actively adopting progressive practices, such as experimenting with novel crop varieties, implementing sustainable farming techniques, and leveraging data-driven approaches to optimize overall production processes.

## 5. Conclusions

This study used a panel data set of cereal farms derived from the FBS in order to assess variations in productivity change in the sector, through the estimation of a MI of TFP. Both $\Delta Eff$ and $\Delta Tech$ indicators were employed, along with their sub-components, to explore further the drivers of innovation in management and innovation through investment in human capital. The MI of TFP revealed significant variation in productivity over the 11-year period, but with a moderate overall improvement over the whole period. One limitation of the index used here, is that it does not account for sequential productivity change and hence future work in the area will need to consider recent methodological developments in this area, for example by O'Donnell et al. [78]. Moreover, although the FBS is a comprehensive and detailed database, it lacks information on specific management practices and decision-making processes that might be used as indicators of innovation in management. Without these, it must be conceded, there is no way of providing data on drivers of innovation detailed enough to inform policy design. In terms of future research, a good starting point would be a study to investigate the possibility of including in the FBS data collection exercise measures for a far wider range of managerial behavior, together with follow-up analysis of the impacts of these varied behaviors on different types of innovation leading to improvements in productivity. A parallel data collection exercise and investigation would be required to gain further insights into the role of technological investments in driving shifts of the efficiency frontier and technological progress. In addition to the lack of managerial behavioral data, FBS lacks data on specific capital investments associated with technological improvements; thus, to further comment on innovation and technological improvements, this gap will need to be filled with more detailed farm-based surveys. Moreover, the FBS sample used for the purposes of this manuscript covers the period between 2003 and 2014. This period, although it covers two significant CAP reforms, fails to capture more recent events (Brexit, COVID-19, and the war in Ukraine). Therefore, future work will need to consider FBS data available for more recent periods.

What can be said, based on these results, is that new agricultural policies will need to focus on ways to capitalize on the existing knowledge and experience of farmers in order to design educational programs that facilitate innovation through management efficiency. Also, it would be beneficial to move away from the traditional 'bigger is better because

costs are spread' mentality, towards a more nuanced approach to achieving productivity gains. To facilitate this, the Government needs to fund research to identify the optimal scale of production for the maximization of productivity gains, for different farm systems. This more nuanced approach is more cognitively challenging, and so farmers will need more support in terms of advisory services and planning tools, and the Government needs to do more to encourage, or incentivize, farmers to become better informed, both concerning current research and the decision-support community.

Recent studies highlight these challenges and opportunities post-Brexit (see e.g., [92,93]). Some emphasize the substantial contribution of CAP direct payments to farm business income and the vulnerability of farms to their removal, underscoring the need for new policies that mitigate these risks and enhance farm productivity [93]. Also, the new challenges and uncertainties faced by arable farming in the UK due to Brexit highlight the need for improved resilience and competitive strategies in the new policy landscape. This study has demonstrated that efficiency improvement is a much more complex issue than is assumed, with multiple dimensions, and that historically, policies and market forces have only stimulated efficiency improvements based on one of these dimensions, i.e., economies of scale. For a number of reasons, there must be limits to the gains available from the use of this approach (and other socially undesirable outcomes)—albeit these are not yet understood. Therefore, more attention needs to be directed to making use of other sources of efficiency improvement. This study has identified the likely drivers of improvements in these efficiency dimensions, albeit that better source data would increase resolution. For example, further studies could explore the impact of accumulated capital and the capacity to expand the scale of operations. Also, to understand the impact of scale efficiency change as a low-tech solution in productivity enhancement. These drivers must be the targets for future policy design and for future academic research.

**Supplementary Materials:** The following supporting information can be downloaded at: https://www.mdpi.com/article/10.3390/agriculture14030409/s1, Supplementary Materials, Section S1. Non-parametric Data Envelopment Analysis (DEA). Table S1. Statistical inference of the MI of TFP over the 11 periods for cereal farms. Table S2. Detailed presentation of the pure efficiency change factor for each individual farm in the sample over the 11-year period. Table S3. Detailed presentation of the scale efficiency change factor for each individual farm in the sample over the 11-year period.

**Author Contributions:** Conceptualization, Y.G.; methodology, Y.G.; software, Y.G. and J.C.-G.; validation, Y.G. and J.C.-G.; formal analysis, Y.G. and J.C.-G.; investigation, Y.G. and J.C.-G.; resources, Y.G.; data curation, Y.G. and J.C.-G.; writing—original draft preparation, Y.G.; writing—review and editing, Y.G., J.C.-G. and P.J.; visualization, Y.G. and J.C.-G.; supervision, Y.G.; project administration, Y.G. All authors have read and agreed to the published version of the manuscript.

**Funding:** This research received no external funding.

**Institutional Review Board Statement:** Not applicable.

**Data Availability Statement:** Restrictions apply to the availability of these data. This research is based on the Farm Business Survey (FBS) for England and Wales. The data are available under strict license from the UK Data Service (UKDS—https://ukdataservice.ac.uk/) and is not open data.

**Acknowledgments:** The authors extend their sincere gratitude to the Farm Business Survey (FBS) and the Rural Business Research (RBR) for their invaluable support throughout the duration of this research project. Their significant contributions in providing access to data, managing data intricacies, and offering technical guidance on variable structures have been instrumental to the success of this study. Furthermore, the authors express their heartfelt appreciation to Paul Wilson from the Rural Business Research and Richard Crane from the Farm Business Survey at the University of Reading. Their continuous support, encouragement, and expertise have played a pivotal role in shaping and enhancing the quality of this research endeavor.

**Conflicts of Interest:** The authors declare no conflicts of interest.

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
