# Peer review of "Linking Entrepreneurship to Productivity: Using a Composite Indicator for Farm-Level Innovation in UK Agriculture with Secondary Data"

_agriculture, doi:10.3390/agriculture14030409_

Round 1

Reviewer 1 Report

Comments and Suggestions for Authors

The reviewed article addresses a highly relevant and contemporary topic. The authors have effectively justified the purpose of the study and identified research gaps. Adequate research methods were employed in the study, and the selection of the research sample and variables is sound. The structure and layout of the study are appropriate.

However, I suggest that the authors consider presenting chapter 1.1 after the methodological section. One of the study objectives is to "... map the variables, specifically management actions, that can be reasonably identified as innovation or as factors increasing the likelihood of innovation within agricultural practices."

In my opinion, it is preferable to introduce the goals first and then proceed with their implementation.

The research results are clear, and the conclusions drawn are directly tied to the conducted research. I commend the authors on producing a very well-written article, and I am pleased with the presented results.

In addition, I recommend that the authors address the limitations of the research conducted. In Table 2, which illustrates changes in efficiency, please specify the period to which these changes apply.

Reviewer 2 Report

Comments and Suggestions for Authors

Dear Authors,

the topic chosen for your article appears to be interesting and, in general, well addressed. It is therefore potentially publishable in this journal. The methodology you adopted can easily be applied and replicated in other countries. However, I think it requires a little further effort to make it truly appealing. In some passages of your manuscript, I suggest simplifying the text to facilitate the reading (there are some repetitions). In general, there are too many subparagraphs, it would be advisable to reduce them in number and make the reading of the text more usable.

I have provided some suggestions below that I hope will improve your work:

-          Remove capital letters from the title of the manuscript. Even the titles of the paragraphs must be written without using the initial capital letter for each word;

-          Please, you must indicate the contact details (affiliation, email) of all authors;

-          It is important to define Acronyms/Abbreviations/Initialisms when they appear for the first time. Please check;

-          Please check spaces;

-          At the end of the introduction there is no indication of how the paper is structured;

-          Line 82: “sector.” must be “sector”;

-          tables and figures should be separated from the text. They are preceded and followed by an empty line. This also applies to the equations in the manuscript;

-          What is the reason for using such old data?

-          I believe that non-market sources of revenue should not be excluded from the analysis. I can agree on the fact that they do not vary in response to changes in the scale of production but they will certainly influence entrepreneurial choices and decisions (having or not having a certain sum of money available will certainly influence the quantities and quality of the production factors to be used in the production process);

-          Please check the editing of equation 3;

-          Conclusions: they lack the description of the limits of the study carried out as well as of the future prospects of the same;

-          Please check the editing of the references...it must be in accordance with the guidelines of the journal (References must be numbered in order of appearance in the text (including table captions and figure legends) and listed individually at the end of the manuscript. In the text, reference numbers should be placed in square brackets [ ]).

Reviewer 3 Report

Comments and Suggestions for Authors

Linking Entrepreneurship to Productivity: Developing a Цomposite Indicator for Farm-Level Innovation in UK Agriculture with Secondary Data

Abstract: The author should redo the Abstract, clearly define the subject, the goal of the research and define the results. It is not necessary to list the stages, it can be done later in the introduction or research methodology. The abstract should state the period to which it refers and whether the research was extended from 2015-2023 or only the impact assessment for 2023.

  16 p. Objectives include a literature review to map innovation variables - reformulate.

104 of heterogeneity in the population of farm businesses being studied – explain.

The Introduction chapter is very confusing and long. It should be divided into Introduction and parts related to productivity, innovation and entrepreneurship. Explain the period of research in particular, emphasizing why?  It is necessary to make a comparison with the data from the next period or using another method to calculate the impact at the end of 2023.

Not equal attention is given to each area. At the end of the Introduction, the structure of the paper should be written. At the end of each sub-chapter (productivity, innovation and entrepreneurship) the literature gap and space that this study fills should be indicated. Make hypotheses. For each area you can create a table of supporting literature (like 154).

210 - Data used in the modeling exercise is sourced from a representative sample of 60, nowhere is it indicated which farms it refers to, how they were selected, which region, country...  explain the representativeness of the sample, what percentage are those 60 farms.

210-218 Cereal farms spanning the years 2003-2014. The author should very carefully explain why this period was used. Although the data for that period were available, it is not clear why the entire subsequent period (from 2015-2023) was not included. Due to the fact that a lot of time and events have passed since 2014, I am of the opinion that the above data is not current. Brexit, the Corona virus pandemic, market-driven consolidation 551p. -  It is necessary to make a comparison with the data from the next period or using another method to project the impact at the end of 2023.

Results

376 -  The statistical analysis of the MI of TFP for the period 2003 and 2014 is detailed in Supplementary Material, Table S1 why the next period is not included - add.

400 Figure 2. Total factor productivity, efficiency, and technical change for the 11-year period - to be supplemented in the following years.

417 - Table 3. The MI of TFP (Malmquist Index) per year and per farm size - to be supplemented in the following years.

698-702... the author lists the conclusions of some studies. That should be moved to the literature section. The author should explain his results and benefits for all services. When additional research is done for the next period, then they will get the right conclusion about innovation and productivity.
